Insights of the dental calculi microbiome of pre-Columbian inhabitants from Puerto Rico

Santiago-Rodriguez Tasha M. tsantiagoro@gmail.com 1 2 3
Narganes-Storde Yvonne 4
Chanlatte-Baik Luis 4
Toranzos Gary A. 5
Cano Raul J. 1 2 3
1 Center for Applications in Biotechnology, California Polytechnic State University—San Luis Obispo , San Luis Obispo , CA , United States of America
2 Biology Deparment, California Polytechnic State University—San Luis Obispo , San Luis Obispo , CA , United States of America
3 Institute for Life Science Entrepreneurship, ATCC-Center for Translational Microbiology , Union , NJ , United States of America
4 Center for Archaeological Investigations, University of Puerto Rico , San Juan , Puerto Rico
5 Biology Department, University of Puerto Rico , San Juan , Puerto Rico
Josenhans Christine
Electronic publication date: 2017 May 2
Publication date: 2017
Volume: 5
Electronic Location ID: e3277
Received 2016 Dec 9; Accepted 2017 Apr 5
Copyright: ©2017 Santiago-Rodriguez et al.
Copyright year: 2017
Copyright holder: Santiago-Rodriguez et al.
License: This is an open access article distributed under the terms of the Creative Commons Attribution License, which permits unrestricted use, distribution, reproduction and adaptation in any medium and for any purpose provided that it is properly attributed. For attribution, the original author(s), title, publication source (PeerJ) and either DOI or URL of the article must be cited.
License URL: https://creativecommons.org/licenses/by/4.0/

Erratum in: Correction: Insights of the dental calculi microbiome of pre-Columbian inhabitants from Puerto Rico 5 6 6 2017 e3277/correction-1 PeerJ PMC5468783 28616368
Keywords: Ancient microbiomes, Bacteria, Saladoid, Oral microbiome, Dental plaque, pre-Columbian cultures, Sorcé

Funding: Howard Hughes Medical Institute This work was supported by a postdoctoral fellowship from the Howard Hughes Medical Institute granted to TS-R through the Life Sciences Research Foundation. The funders had no role in study design, data collection and analysis, decision to publish, or preparation of the manuscript.

==============================
Background

The study of ancient microorganisms in mineralized dental plaque or calculi is providing insights into microbial evolution, as well as lifestyles and disease states of extinct cultures; yet, little is still known about the oral microbial community structure and function of pre-Columbian Caribbean cultures. In the present study, we investigated the dental calculi microbiome and predicted function of one of these cultures, known as the Saladoid. The Saladoids were horticulturalists that emphasized root-crop production. Fruits, as well as small marine and terrestrial animals were also part of the Saladoid diet.

Methods

Dental calculi samples were recovered from the archaeological site of Sorcé, in the municipal island of Vieques, Puerto Rico, characterized using 16S rRNA gene high-throughput sequencing, and compared to the microbiome of previously characterized coprolites of the same culture, as well modern plaque, saliva and stool microbiomes available from the Human Microbiome Project.

Results

Actinobacteria, Proteobacteria and Firmicutes comprised the majority of the Saladoid dental calculi microbiome. The Saladoid dental calculi microbiome was distinct when compared to those of modern saliva and dental plaque, but showed the presence of common inhabitants of modern oral cavities including Streptococcus sp., Veillonella dispar and Rothia mucilaginosa. Cell motility, signal transduction and biosynthesis of other secondary metabolites may be unique features of the Saladoid microbiome.

Discussion

Results suggest that the Saladoid dental calculi microbiome structure and function may possibly reflect a horticulturalist lifestyle and distinct dietary habits. Results also open the opportunity to further elucidate oral disease states in extinct Caribbean cultures and extinct indigenous cultures with similar lifestyles.

Introduction

Paleomicrobiology or the study of ancient microorganisms is providing insights into microbial evolution, the lifestyles of extinct cultures, as well as ancient diseases. Mummified gut remains (Ubaldi et al., 1998; Cano et al., 2000; Santiago-Rodriguez et al., 2015), coprolites (Tito et al., 2012; Santiago-Rodriguez et al., 2013; Cano et al., 2014), bones (Yang et al., 1998) and dental calculi (Adler et al., 2013; Warinner et al., 2014; Warinner, Speller & Collins, 2015; Ziesemer et al., 2015) are known to harbor ancient microbial DNA. Dental calculi, particularly, is known to be commonly recovered from archaeological excavations across diverse geographical regions, and studies have shown that it is an important bacterial reservoir (Vandermeersch et al., 1994; Arensburg, 1996; Preus et al., 2011). Previous paleomicrobiology studies relied heavily on techniques such as microscopy and amplification of genes from particular bacterial species using the Polymerase Chain Reaction (PCR) (Pääbo, Higuchi & Wilson, 1989; Cano & Borucki, 1995; Pap et al., 1995; Willerslev & Cooper, 2005). Recent advances in high-throughput sequencing and bioinfomatic tools have enabled the study of bacterial communities by simultaneously amplifying and sequencing the 16S ribosomal RNA (16S rRNA) gene (Caporaso et al., 2010).

The modern human oral microbiome includes diverse communities composed of >1,000 bacterial species belonging to the Actinobacteria, Bacteroidetes, Firmicutes and Proteobacteria (Dewhirst et al., 2010). Unlike the human gut microbiome, little is still known about the potential effect that lifestyles, dietary habits and oral hygiene may exert on the oral microbiome (Adler et al., 2013; Clemente et al., 2015). A previous study characterizing the oral microbiome of individuals from three distinct geographical regions found profound bacterial taxonomic differences that may be attributed to culture (Li et al., 2014). Another study found that the oral microbiome of uncontacted Amerindians is distinct and more diverse compared to subjects with westernized lifestyles (Clemente et al., 2015). Similarly, a study characterizing the gut microbiome of hunter-gatherers from South America showed that it is different and more diverse compared to that of westernized cultures (Obregon-Tito et al., 2015). These studies indicate that modern, isolated societies have distinct microbiomes compared to modern westernized societies, and that these may better resemble the ancestral or original state of the human microbiome.

Microbial characterization of ancient human societies may potentially provide direct insights into the ancestral state of the oral microbiome (Warinner, Speller & Collins, 2015). Ancient human oral microbiomes have been studied through dental calculi, which have been found to be dominated by Actinobacteria, Firmicutes, Proteobacteria and Bacteroidetes, and also harbor oral pathogens (Warinner et al., 2014). Previous studies have characterized the oral microbiome of ancient European cultures (Adler et al., 2013; Warinner et al., 2014). Results have shown that changes in dietary habits associated with specific time periods affect the structure and function of the human oral microbiome (Adler et al., 2013). Pre-Columbian cultures also had a significant impact on modern western societies, and characterization of their oral microbiome may also provide insights of the ancestral state of the human oral microbiome throughout part of human history (Ziesemer et al., 2015). It is feasible to hypothesize that the recovered dental calculi may resemble taxonomic and functional profiles of modern dental plaque, and also possibly reflect pre-Columbian lifestyles.

Figure 1 Location of samples.

Location and example of the samples obtained. (A) shows the organization of the Caribbean as a site for ancient human migrations. Vieques Island, which is part of the main island of Puerto Rico, is highlighted by the red circle. (B) shows the location of the archaeological site of Sorcé on Vieques. (C) shows examples of the teeth recovered from the archeological site of Sorcé.

Archaeological evidence suggests that pre-Columbian cultures came in close contact with each other, suggesting long-distance migrations possibly due to knowledge in seafaring (Curet, 2005). The arrangement of the Caribbean archipelago may have facilitated the migration of pre-Columbian cultures from diverse regions across the Americas (Fig. 1A). Migration throughout the Caribbean resulted in the settlement and development of a diverse mosaic of cultures. Puerto Rico, particularly, was an important site of ancient human migrations in the Caribbean, representing the longest continuous span of occupation in the Antilles (Curet, 2005). Archaeological evidence in Puerto Rico suggest two major pre-Columbian migrations, that included the Archaic people and the Saladoids (Siegel et al., 2005; Siegel, 2010). The Saladoids migrated from South America (lower Orinoco River) and established at the site of Sorcé in the municipal island of Vieques, Puerto Rico around 180 B.C., and the main island of Puerto Rico by 430 B.C. While a major hypothesis suggests that the Saladoid culture was divided in two subseries, the Huecan and the Cedrosan Saladoid, an alternate hypothesis suggests that each of the subseries were unique and distinct cultures (Siegel et al., 2005; Ramos, 2010). The alternate hypothesis is mainly supported by differences in semi-precious stone ornaments, and ceramic paintings and decorations, the latter of which still remain the primary analytical units of pre-Columbian Caribbean cultures. Based on the alternative hypothesis, the Saladoids co-inhabited with the Huecoids in Sorcé for over 1,000 years (5 A.D.–1170 A.D.) (Siegel, 2010). Archaeological and isotope evidence have suggested that the Saladoids relied on terrestrial resources and brought to Puerto Rico a developed horticultural economy, which may have been modified with interactions with Archaic groups (Siegel et al., 2005). The Saladoid diet included manioc or yuca, sweet potatoes, yautia, squashes, maize, nuts, as well as fruits such as soursop, papaya, guava and passion fruit (Siegel et al., 2005). Proteins were mainly acquired from the ingestion of fish, crustaceans, bivalves, sea turtles, birds and small terrestrial mammals (Storde, 1982; Baik, 2013). While food preparation methods remain largely unknown (Siegel et al., 2005), thoroughly boiling food items, such as tubers, prior to consumption would have been a feasible practice. Food boiling may be consistent with the absence of nucleic acids from food items, such as manioc, in ancient human samples.

Recently, microbial community analyses of coprolites found in Saladoid deposits from Sorcé have shown marked differences when compared to coprolites found in Huecoid deposits from the same site, as well as modern Amazonian cultures, suggesting that the gut microbiome may reflect culture-specific dietary habits (Santiago-Rodriguez et al., 2013; Cano et al., 2014). Teeth with dental calculi from the Saladoid culture at Sorcé have also been recovered, representing an opportunity to provide insights into their oral microbiome. Unfortunately, Huecoid human remains, including teeth, have not been recovered from the archaeological site of Sorcé and were not included in the present study; yet, the previously characterized coprolite microbiomes of the Saladoid culture, as well as the microbiomes of supragingival and subginvival plaque, saliva, stool and soil from the archaeological site of Sorcé have been included in the present study. It is hypothesized that the oral microbiome of the Saladoids may be distinct from modern individuals, possibly due to differences in culture, lifestyles, dietary habits, and lack of oral hygiene; therefore, the aims of the present study were to characterize the taxonomic and predicted functional categories of Saladoid dental calculi using 16S rRNA gene high-throughput sequencing, and compare them to the microbiomes of modern oral, gut and soil from the archaeological site of Sorcé.

Table 1 Sample description.

All samples were collected from the archeological site YTA-2 at the Sorcé Estate in La Hueca, Vieques, Puerto Rico. Samples date from 10 A.D. to 385 A.D.

Sample	Depth (m)	Description	Weight (mg)	Sequences prior filtering	OTUs prior filtering	Sequences after filtering	OTUs after filtering	
D21	0.60–0.80	Maxillary alveolar bone fragment with (6) teeth. Moderate wear.	2.50	2,677	497	799	304	
E19	0.20–0.40	Loose tooth with severe wear.	2.78	4,309	571	1,297	371	
E26	0.60	Loose molar tooth with severe wear.	3.02	5,933	490	604	288	
F20	0.80	Loose tooth with moderate wear.	4.10	3,345	334	337	178	
G18	0.40–0.60	Loose teeth (2) with wear	3.89	14,303	363	374	196	
G21	0.20–0.40	Loose molar tooth with abscess, and teeth (3). Moderate wear.	2.92	24,371	480	879	300	
G22	0.20–0.40	Loose tooth with enamel detachment. Moderate wear.	4.35	5,227	707	1,863	505	
I19	0.00–0.20	Loose teeth (2). Moderate wear.	2.37	3,523	499	877	339	
I23A	0.20–0.40	Molar tooth and tooth. Moderate wear.	4.28	2,424	495	673	308	
I23B	0.40–0.60	Maxillary alveolar bone fragment with (2) teeth and loose tooth. Moderate wear.	3.17	2,031	439	602	278	
I24A	0.20–0.40	Loose tooth. Moderate wear.	1.95	2,125	418	625	265	
I24B	0.60–0.80	Maxillary alveolar bone fragment with (3) teeth and (2) loose teeth. Moderate wear.	2.02	2,321	376	602	228	
F24	0.60	Mandibular alveolar bone fragment with attached teeth (11). Male. Moderate wear.	5.25	3,005	110	245	33	
H6	0.70	Infant maxillary alveolar bone fragment with fully erupted tooth and deciduous teeth (3)	5.14	56,503	74	334	31	
M8	0.00–0.20	Mandibular alveolar bone fragment with (4) teeth. Male. Moderate wear	4.98	2,214	46	267	23	
Soil*	 	Archaeological site of Sorcé	 –	22,945	336	–	–	
Blank*			–	34,392	956	–	–	
Notes.

Dental calculi samples highlighted in bold were attached to bone fragments that enabled the gender or approximate age determination.

* Soil and blank sequences were filtered from calculi samples for main analyses.

Materials and Methods

Archaeological site and sample description

The archaeological excavation site was located at the Sorcé Estate in La Hueca, Vieques, Puerto Rico (Fig. 1B). The excavation was directed by archeologists Luis Chanlatte-Baik and Yvonne Narganes-Storde from the Center for Archaeological Investigations at the University of Puerto Rico. A total of 12 loose dental samples were included in the present study and were excavated from the Saladoid deposit YTA-2 at different depths. Three dental calculi samples were recovered from teeth attached to bone fragments that enabled gender or age determination by forensic archeologist Dr. Edwin Crespo from the University of Puerto Rico (Table 1). Age at death and sex were determined by the application of morphological and quantitative methods as described in Bass (1971), Krogman & Işcan (1986), and Scheuer & Black (2000). Teeth for this study were acquired with all necessary field permits complying with all relevant regulations for the collection of skeletal remains from archeological sites. Repository information, including the nomenclature for precise identification containing geographical location, excavation site and archaeological depth are described in Table 1. 14C dating was conducted by Teledyne Isotopes (Westwood, NJ) and BETA Analytic, Inc. (Miami, FL, USA) using standard methods. Most recovered teeth samples correspond to adult subjects, with the exception of sample H6, which was identified as an infant, showing moderate to severe wear and carious lesions. Figure 1C shows examples of the teeth included in the present study.

Dental calculi recovery and DNA extraction

Dental calculi recovery was performed in a class II hood at the Center for Applications in Biotechnology at the California Polytechnic State University, San Luis Obispo, dedicated for ancient DNA analyses. Hoods were sterilized with UV-radiation at least 15 min before and after every use. Dental calculi were separated from teeth using a dental scalar kindly provided by Dr. Rodney Hiltbrandt, D.D.S. that was sterilized before and after every use with UV-radiation and bleach between samples. Protective clothing and gloves were worn at all times, and were also changed between samples. The recovered dental calculi were then placed in sterilized tubes and pulverized using custom-made micropestles. Micropestles were sterilized using UV-radiation and bleach between samples. A mixture of supragingival and subgingival calculi was obtained from each tooth as it was not possible to make a clear distinction. In order to reduce environmental contamination, DNA extraction was performed in dedicated class II hood, where the calculi was also recovered, using the PowerSoil DNA Isolation Kit following manufacturer’s instructions with the following modifications: (i) the PowerBead tube was placed in a Qbiogene Fast Prep Instrument (Carlsbad, CA) at 4.5 m/s for 30 s and then centrifuged at 10,000× g for 30 s; and (ii) incubation periods at −20 °C for 15 min (Mo Bio Laboratories, Carlsbad, CA, USA). The PowerSoil DNA Isolation Kit can recover DNA fragments as short as 60 bp (Qiagen technical support, pers. comm., 2017). All necessary equipment for DNA extraction, including micropipets, was sterilized before and after every use with a 10% bleach solution and 70% ethanol. Soil from the archaeological site of Sorcé surrounding the maxillary fragment of sample M8 (Table 1) was recovered as a control for further analyses. A blank control was also included throughout the DNA extraction, PCR amplification and sequencing as described below.

16S rRNA gene amplification and sequencing

DNA amplification of the 16S rRNA gene was performed at Molecular Research Laboratory (MRDNA) (http://www.mrdnalab.com; Shallowater, TX, USA). All DNA samples were handled in exclusive areas for PCR amplification, which are sterilized before and after every use using DNAaway and UV-radiation to eliminate cross-contamination with modern samples. Template manipulations are handled in separate hoods that are sterilized before and after every manipulation using DNAaway and UV-radiation. Negative PCR controls were included in all amplification reactions, showing no amplification. The 16S rRNA gene V4 variable region was amplified using the universal PCR primers 515f (GTGCCAGCMGCCGCGGTAA)/806r (GGACTACHVGGGTWTCTAAT), as described previously (Caporaso et al., 2012). PCR amplifications were conducted using a single step 30 cycle PCR using the HotStarTaq Plus Master Mix Kit (Qiagen, USA) under the following conditions: 94 °C for 3 min, followed by 28 cycles of 94 °C for 30 s, 53 °C for 40 s and 72 °C for 1 min, after which a final elongation step at 72 °C for 5 min was performed. Two PCR amplifications following the conditions described above were performed due to the limiting starting material and to obtain higher yields to continue with library preparation. After amplification, PCR products were checked in 2% agarose gel to determine the success of amplification and the relative intensity of the bands. All amplicon products from each sample were mixed or pooled after tagmentation in equal concentrations and purified using Agencourt AMPure beads (Agencourt Bioscience Corporation, MA, USA). The pooled and purified PCR products were used to prepare the DNA library following Illumina MiSeq DNA library preparation protocol using the MiSeq reagent kit v2 (2×150 bp) on a MiSeq following the manufacturer’s guidelines. Preprocessing of reads including removal of chimeras and singletons was performed using proprietary tools from MRDNA. Phred scores were checked using the FastQC tool http://www.bioinformatics.babraham.ac.uk/projects/fastqc/.

16S rRNA gene analyses

Due to data availability, modern oral and gut samples used for comparison were obtained from the Human Microbiome Project (HMP) (http:/hmpdacc.org/). Data correspond to overall healthy individuals, age 18–40, from the USA. Further information about these subjects is available at http://www.ncbi.nlm.nih.gov/projects/gap/cgi-bin/study.cgi?study_id=phs000228.v3.p1. Stool samples corresponded to an Amazonian culture. These samples were chosen over USA individuals because that may better resemble that of the Saladoid culture due to similarities in lifestyles. All 16S rRNA gene sequence analyses were performed using the Quantitative Insights Into Microbial Ecology (Qiime) (Caporaso et al., 2010). Reads were assigned to samples based on their corresponding barcode using split_libraries.py with default filtering parameters. Operational taxonomic units (OTUs) were selected using the pick_closed_reference_otus.py workflow in Qiime because different regions of the 16S rRNA gene were being characterized. 16S rRNA taxonomy was defined by 97% similarity to reference sequences. OTU biom files were created for the modern and ancient oral and gut microbiome sequences, and checked using biom summarize-table. OTU biom files were also created separately for the blank control and soil from the archeological site of Sorcé, and checked using biom summarize-table. OTUs present in the blank control (Data S1) and soil from the archaeological site of Sorcé (Data S2) were removed from all the samples using the workflow described in http://qiime.org/tutorials/filtering_contamination_otus.html to eliminate the potential contamination effect of laboratory reagents and post-mortem environment to the results. All subsequent analyses were also performed with unfiltered soil and blank control OTUs.

Bayesian Source Tracker or SourceTracker analyses were then performed to identify possible sources of contamination and/or microbiomes sharing some resemblance with the dental calculi microbiomes (Knights et al., 2011). Microbiomes included in the Bayesian Source Tracker analyses were supragingival plaque (n = 5), subgingival plaque (n = 5), saliva (n = 5), stool (n = 5), coprolite (n = 5) (described below), soil from the archaeological site of Sorcé (n = 1), and a blank control (n = 1). Coprolite sequences were included, since certain microorganisms may be part of both oral and gut microbiomes. The mentioned ancient and modern oral, as well as gut microbiomes were also considered in further analyses (Table S1).

The lowest sequence count was 245 after removing from the dental calculi samples OTUs corresponding to the blank control and soil from the archaeological site of Sorcé; thus, data were rarefied to 245 sequences to minimize the effect of disparate sequence number to the results. Rarefaction allows the calculation of species richness through the construction of rarefaction curves. Taxonomy, as well as alpha- and beta-diversity plots were determined using the core_diversity_analyses.py workflow. Alpha diversities, including chao1 and observed OTUs were computed from the average of ten iterations using the collate_alpha.py workflow (Chao, 1984; Gotelli & Colwell, 2001). Statistical analyses were performed using the alpha_compare.py workflow with default parameters. Since subsampled reads for each comparison were used, a Bonferroni correction was applied. 2D plots were constructed as described http://qiime.org/scripts/make_2d_plots.html. Group significance analyses were performed using the group_signficance.py script in Qiime with default parameters, including Kruskal–Wallis as the statistical test and 1,000 permutations. The test compares OTU frequencies between sample groups to see if these are differentially represented. The category considered in this analysis was sample type that included dental calculi, dental calculi from teeth attached to bone, supragingival plaque, subgingival plaque, saliva, stool, coprolite and soil from the archaeological site of Sorcé. The core microbiome between the dental calculi and modern oral microbiomes (i.e., supragingival and subgingival plaque, as well as saliva) was computed using the compute_core_microbiome.py after filtering soil and blank OTUs in order to identify shared OTUs between ancient and modern oral samples. The script identifies the core OTUs in a biom table. Minimum fraction for core was set to 0.25 or 25%. Analyses stopped at 0.40 or 40%, indicating that no OTUs were shared in over 40% of the samples.

Prediction of functional categories based on 16S rRNA gene data was performed using the package Phylogenetic Investigation of Communities by Reconstruction of Unobserved States (PICRUSt) available online at https://huttenhower.sph.harvard.edu/galaxy/ (Langille et al., 2013). Functional profiles predicted at level 2 were visualized using a heatmap that was constructed using the function heatmap.2 available in the R package gplots. The relative abundances of the functional profiles were also visualized using Linear Discriminatory Analysis Effect size (LEfSe) plots. LEfSe determines the features most likely to explain differences between categories (e.g., sample type). Analyses were also performed with unfiltered soil and blank control OTUs.

Results

16S rRNA gene diversity of dental calculi

Fifteen dental calculi samples from the Saladoid culture were recovered from the archaeological site of Sorcé, Puerto Rico, and the DNA characterized using 16S rRNA gene high-throughput sequencing (Table 1). An average of 692 ± 429 dental calculi sequences with an average size of 272 bp (after quality filtering including chimeras and singletons removal) were analyzed. Average Phred score was 40. An average of 5,097 ± 1,007 (supragingival plaque); 4,094 ± 372 (subgingival plaque); 5,686 ± 1369 (saliva); 2,640,130 ± 548,433 (stool); 48,100 ± 32,461 (coprolites); and 22,945 (soil from the archaeological site of Sorcé) sequences were also analyzed. OTUs corresponding to a blank control and soil from the archaeological site of Sorcé are included in Data S1 and S2 respectively, and were filtered from the dental calculi OTUs, as described above. Prior any further analyses, we performed SourceTracker analyses to identify possible sources of contamination, and modern microbiomes that could potentially share resemblance with the dental calculi microbiomes. Results showed that >99.0% of the dental calculi sequences did not resemble any of the modern or ancient human microbiomes (shown as unknown in grey) (Figs. S1A–S1O). SourceTracker analyses were also performed prior to filtering soil and blank control OTUs. Results showed a percentage of OTUs in the dental calculi samples that resemble soil and blank controls prior to filtering (Figs. S1A–S1O). The dental calculi microbiomes were then compared to those of coprolites from the same culture, and modern dental plaque, saliva, stool and soil from the archaeological site of Sorcé. Data were rarefied to 245 sequences (given that this was the lowest sequence count after filtering OTUs from soil and blank control) in order to minimize the effect of disparate sequence number on the results (Table 1 and Table S1). Samples reached or were close to reaching a plateau when data were rarefied, as demonstrated with the chao1 (Fig. S2A) and observed OTUs (Fig. S2B) rarefaction plots. Rarefaction curves of chao1 (Fig. S2C) and observed OTUs (Fig. S2D) indices prior to filtering OTUs corresponding to soil and blank control from the dental calculi samples are also shown. Two different alpha diversity indices including chao1 (Fig. 2A) and observed OTUs (Fig. 2B) were plotted after filtering OTUs corresponding to soil and blank control. Chao1 (Fig. S3A) and observed OTUs (Fig. S3B) indices prior to filtering soil and blank control OTUs are also shown. Chao1 (Table S2) and observed OTUs (Table S3) values exhibiting statistically significant differences are highlighted in bold. Values shown in Tables S2 and S3 correspond to samples after filtering soil and blank control OTUs. Analyses were also performed with unfiltered soil and blank control OTUs, showing differing p-values in both chao1 (Table S4) and observed OTUs (Table S5) when compared to filtered soil and blank control OTUs.

Figure 2 Alpha Diversity Filtered.

Bar plots of alpha diversity indices. Bar plots representing the chao 1 (A) and observed OTUs (B) indices for the bacterial taxonomy based on 16S rRNA gene of the dental calculi, modern supragingival and subgingival plaque, saliva, coprolites, stool and soil from the archaeological site of Sorcé microbiomes. Alpha diversity indices were computed from the average of ten iterations using the collate_alpha.py workflow. Soil from the archeological site of Sorcé and blank control OTUs were filtered from the dental calculi prior analyses.

Figure 3 PCoA Filtered.

Principal Coordinates Analysis (PCoA) 2D plots of ancient and modern oral and gut microbiomes, as well as that of soil from the archaeological site of Sorcé. Dental calculi, dental calculi of teeth attached to bones that enabled the identification of gender or age (Dental calculi (Bone)), coprolites, stool, soil from the archaeological site of Sorcé, supragingival (grey) and subgingival plaque (black), and saliva (white). Soil from the archeological site of Sorcé and blank control OTUs were filtered from the dental calculi prior analyses.

Principal Coordinate Analyses (PCoA) plots were constructed to visualize beta-diversity indices of the modern and ancient oral and gut microbiomes, as well as that of soil from the archaeological site of Sorcé (Fig. 3). Results show clustering based on sample type. Analyses were also performed with unfiltered soil and blank control OTUs, showing a separation based on sample type. Interestingly, the soil sample clustered closely to stool samples from Amazonians (Fig. S4). On average, Actinobacteria (14.7%), Proteobacteria (28.4%) and Firmicutes (26.5%) comprised the majority of the Saladoid dental calculi microbiome at the phylum level. Notably, the taxonomic composition of the dental calculi recovered from bone that enabled gender or age determination exhibited variations in these relative abundances. For instance, sample H6, identified as an infant, had 55.7% of the OTUs classified as Bacteroidetes, from which 43.2% were Bacteroides sp. Actinobacteria comprised an average of 29.4%, 12.7%, 2.6%, 2.3% and 63.0% of the bacterial communities in modern supragingival and subgingival plaque, saliva and stool, and coprolites, respectively (Fig. 4). The majority of the modern supragingival plaque microbiome was comprised by Firmicutes (27.3%), Bacteroidetes (17.6%), Fusobacteria (13.5%) and Proteobacteria (10.3%). The majority of the modern subgingival plaque microbiome was also composed of the same phyla as supragingival plaque, but in differing proportions, depending on the subject (Fig. 4). Notably, the microbiome of soil from the archaeological site of Sorcé was mostly comprised by Firmicutes (57.3%) and Actinobacteria (34.8%) (Fig. 4). Taxa identified in the blank control are shown in Data S3. Taxonomy analyses at the phylum level with unfiltered soil and blank control OTUs showed differing proportions of Proteobacteria and Firmicutes (Fig. S5). Taxonomic analyses at the genus level after filtering soil and blank control OTUs enabled the identification of 869 taxonomic groups, where 345 could not be classified at the genus level (Data S4). Genera including Corynebacterium sp., Lactobacillus sp., Rothia sp., Staphylococcus sp., Streptococcus sp., and Treponema sp. were identified in both ancient and modern oral microbiomes. Taxonomic classification at the genus level prior filtering soil and blank OTUs showed 840 taxonomic groups, where 325 could not be classified at the genus level (Data S5). A number of unclassified taxonomic groups that were present in soil were also identified in both ancient and modern microbiomes.

Group significance analyses after filtering soil and blank control OTUs showed a number of unique OTUs in the dental calculi microbiomes that were not identified in the modern and ancient gut or oral microbiomes (Data S6). There were also a number of shared OTUs between modern oral and dental calculi microbiomes including, but not limited to Streptococcus (OTU 1082294), Veillonella dispar (OTU 962249), Rothia mucilaginosa (OTU 1017181) and Porphyromonas sp. (OTU 4301737) (Table 2 and Data S6). Group significance analyses prior to filtering soil and blank OTUs showed OTUs shared between soil and the dental calculi samples (Data S7). Interestingly, OTUs 172063 (Bacillus sp.), 961009 (Acinetobacter johnsonii), 1088265 (Propionibacterium acnes) and 3453734 (Acinetobacter sp.) were identified in the soil, dental calculi and Amazonian stool samples. Core microbiome analyses of the dental calculi samples after filtering soil and blank OTUs, supragingival and subgingival plaque, as well as saliva showed Rothia mucilaginosa in both ancient (Data S8) and modern oral (Data S9) microbiomes.

Figure 4 Taxa Plots Filtered.

Barplots representing the bacterial taxonomy based on 16S rRNA gene. Data are shown at the phylum level for dental calculi, supragingival plaque, subgingival plaque, saliva, coprolites, stool and soil from the archeological site of Sorcé. Dental calculi samples attached to bone that enabled age and gender determination are highlighted in red. Soil from the archeological site of Sorcé and blank control OTUs were filtered from the dental calculi prior analyses.

Table 2 Group significance analyses of dental calculi, dental calculi (Bone), coprolites, supragingival and subgingival plaque, saliva, stool and soil from the archaeological site of Sorcé.

Selected OTUs present in both dental calculus and modern oral samples are shown. Group significance analyses were performed after filtering soil and blank OTUs.

OTU	Test-Statistic	P	FDRP	Bonferroni P	Soil*	Saliva*	Coprolite*	Calculi*	Subgingival plaque*	Calculi (Bone)*	Supragingival plaque*	Stool*	Taxonomy	
349024	29.790	0.000	0.000	1.000	0	0.000	0	0.833	0.000	0.000	0.250	32427.600	Streptococcus sp.	
1090059	26.416	0.000	0.002	1.000	0	3.500	0	0.250	1.000	0.000	2.750	174.400	Granulicatella sp.	
139056	5.396	0.612	0.735	1.000	0	0.250	0	0.250	0.250	0.000	0.000	0.000	Aggregatibacter sp.	
966331	5.396	0.612	0.735	1.000	0	0.000	0	0.250	0.250	0.000	0.250	0.000	Dermacoccus sp.	
1082294	36.401	0.000	0.000	0.087	0	72.750	0	0.167	18.250	0.000	207.000	0.000	Streptococcus sp.	
1065974	22.882	0.002	0.005	1.000	0	5.250	0	0.167	1.500	0.000	4.500	46.800	Vagococcus sp.	
131775	5.464	0.603	0.725	1.000	0	0.000	0	0.167	0.000	0.000	1.250	0.000	Micrococcaceae sp.	
535196	5.123	0.645	0.759	1.000	0	0.000	0	0.167	0.250	0.000	0.000	0.000	Rhodobacteraceae sp.	
962249	34.888	0.000	0.000	0.167	0	18.500	0	0.083	12.250	0.000	12.250	0.000	Veillonella dispar	
1017181	29.704	0.000	0.000	1.000	0	8.750	0	0.083	4.500	0.000	4.000	20.400	Rothia mucilaginosa	
1059729	28.524	0.000	0.001	1.000	0	6.500	0	0.083	1.000	0.000	5.000	4.400	Granulicatella sp.	
73471	22.635	0.002	0.006	1.000	0	0.000	0	0.083	7.750	0.000	16.250	1.200	Actinomyces sp.	
1084417	19.585	0.007	0.018	1.000	0	5.000	0	0.083	12.000	0.000	9.250	2.800	Lautropia sp.	
925707	15.381	0.031	0.065	1.000	0	1.000	0	0.083	12.750	0.000	0.750	2.000	Gemellaceae sp.	
499378	14.442	0.044	0.090	1.000	0	0.000	0	0.083	0.500	0.000	0.750	0.000	Actinomyces sp.	
509773	10.181	0.179	0.334	1.000	0	18.500	0	0.083	2.000	0.000	0.250	0.000	Streptococcus sp.	
577170	9.319	0.231	0.398	1.000	0	0.000	0	0.083	0.250	0.000	0.000	4.400	Bacteroides sp.	
542066	5.629	0.584	0.702	1.000	0	0.250	0	0.083	0.000	0.000	0.250	0.000	Actinomycetales sp.	
432284	5.464	0.603	0.725	1.000	0	0.500	0	0.083	0.000	0.000	0.000	0.000	Streptophyta sp.	
1005406	5.293	0.624	0.736	1.000	0	0.250	0	0.083	0.000	0.000	0.000	0.000	Staphylococcus aureus	
898207	5.293	0.624	0.736	1.000	0	0.250	0	0.083	0.000	0.000	0.000	0.000	Granulicatella sp.	
336228	5.293	0.624	0.736	1.000	0	0.250	0	0.083	0.000	0.000	0.000	0.000	Dialister sp.	
4301737	19.335	0.007	0.020	1.000	0	0.500	0	0.000	0.000	13.000	0.250	0.000	Porphyromonas sp.	
341460	35.743	0.000	0.000	0.115	0	180.750	0	0.000	29.500	0.667	40.500	0.000	Haemophilus sp.	
530164	20.981	0.004	0.011	1.000	0	102.750	0	0.000	41.000	0.667	1.000	0.000	Porphyromonas sp.	
Notes.

* Sample means are shown.

Predicted functional categories

Functional categories in ancient and modern oral and gut microbiomes were predicted using PICRUSt. In principle, PICRUSt does not predict functional categories from 16S rRNA gene data, since the originating organisms are not being detected, but rather the “bleed over” of putative pathways from the underlying catalog due to cross-annotation of gene families, computationally detected sequence homology, or very rarely from sequence misclassification. Most often this is due to conserved genes being annotated to those pathways in KEGG and other databases, although the actual underlying gene family is something more basic (e.g., cell cycle or structural) (C Huttenhower, pers. comm., 2015). The relative abundances of the functional categories at level 2 were visualized using heatmap plots (Fig. 5). Overall, the modern gut and oral microbiomes clustered separately from the coprolites, dental calculi and soil from the archaeological site of Sorcé (Fig. 5). In the cluster including the modern microbiomes, there was a separation based on biogeographical site. In the cluster including the ancient microbiomes, there was, in turn, a separation between the coprolites and dental calculi microbiomes (Fig. 5). Functional categories were also predicted prior to filtering soil and blank OTUs. Results also showed a separation of the dental calculi samples from the modern microbiomes, but expectedly, functional profiles exhibited differing proportions (Fig. S6).

Figure 5 PICRUSt.

Heatmap of the normalized relative abundances of the predicted functional categories (level 2) of the microbiomes of dental calculi, coprolites, supragingival and subgingival plaque, saliva, stool and soil from the archeological site of Sorcé. Functional categories were predicted using PICRUSt. Dental calculi samples attached to bone that enabled age and gender determination are highlighted in red. Soil from the archeological site of Sorcé and blank control OTUs were filtered prior analyses.

Differences in the functional categories predicted using PICRUSt were visualized using LEfSe plots. Dental calculi microbiomes were characterized by functional categories associated with cell motility, signal transduction and biosynthesis of other secondary metabolites. Modern oral microbiomes were characterized by functional categories associated with replication and repair; translation; nucleotide metabolism; metabolism of cofactors and vitamins; glycan biosynthesis and metabolism; folding, sorting and degradation; cell growth and death; as well as signaling molecules and interaction. The microbiome of soil from the archaeological site of Sorcé was characterized by metabolism of other amino acids (Fig. 6). Unique functional categories of the coprolites and modern gut microbiomes are also shown in Fig. 6. LEfSe analyses of unfiltered soil and blank control OTUs showed that cell motility and signal transduction were still unique features of the dental calculi samples. Categories unique to each sample type were similar prior to and after filtering soil and blank control OTUs (Fig. S7).

Figure 6 LEfSe Filtered.

Linear discriminatory analyses. Effect size (LEfSe) plots of predicted functional categories (level 2). Functional categories of dental calculi (ancient oral microbiomes), supragingival plaque, subgingival plaque and saliva (modern oral microbiomes), coprolites (ancient gut microbiomes), stool (modern gut microbiomes) and soil from the archeological site of Sorcé (tropical soil) were predicted using PICRUSt. Soil from the archeological site of Sorcé and blank control OTUs were filtered prior analyses.

Discussion

Contamination of human samples by the post-mortem environment is an issue when working with ancient oral samples that have been in direct contact with soil as microbes may penetrate the dentition. This issue has previously been raised for dental calculi found in archaeological sites in the Virgin Islands (Ziesemer et al., 2015). One possible approach to authenticate ancient microbiomes is utilizing Bayesian Source Tracker analyses (Tito et al., 2012; Warinner et al., 2014; Santiago-Rodriguez et al., 2015). In the present study, the Saladoid’s dental calculi microbiomes did not resemble the modern or ancient microbiomes included in the analyses. SourceTracker analyses showed a number of OTUs in the dental calculi samples resembling soil from the archaeological site of Sorcé and a blank control; thus, these OTUs were filtered from the dental calculi sequences prior main analyses. Filtering OTUs corresponding to soil from the archaeological site of Sorcé and a blank control possibly increased the potential of analyzing sequences associated exclusively with dental calculi. A previous study showed that even after decontamination of the exterior of dental calculi, a number of soil sequences may still be detected; therefore, filtering soil OTUs may give a better resolution of the authentic dental calculi microbiome (Ziesemer et al., 2015). The present as well as previous studies have found bacterial DNA in commercially available extraction kits (Glassing et al., 2016); therefore, it is of importance to include a blank control in sequencing reactions that could be used to filter potential contaminating OTUs prior further analyses.

While a previous study characterizing the dental calculi microbiomes of ancient European cultures showed that these resembled dental plaque microbiomes of people of European ancestry (Warinner et al., 2014), our data showed that the dental calculi microbiome of the Saladoid culture is taxonomically different at the phylum level and generally more diverse compared to modern dental plaque. It has been argued that this is due to contamination with soil microbes (Warinner et al., 2014). While this is a possibility that is often visited by other researchers in the field of ancient DNA, it should also be considered that differences in dietary habits and hygiene practices may have an impact on the taxonomic structure of the human oral microbiome. Studies characterizing the dental plaque microbiome of modern isolated and industrialized cultures with differences in ancestry, dietary habits and oral hygiene are needed to provide better insights of the evolution of the human oral microbiome throughout part of human history. For instance, a recent study characterizing the saliva microbiome of the Tsimane, an indigenous people from the Amazon, showed that the Firmicutes, Bacteroidetes, Actinobacteria and Proteobacteria comprised the majority of the bacterial taxa in adults. The salivary microbiome of Tsimane children was characterized by not resembling that of their mothers and having a higher relative proportion of Firmicutes (Han et al., 2016). Although researchers sampled saliva, results resemble some our findings. This resemblance may be due to the Tsimane diets consisting primarily of cultivated carbohydrates and fish, similar to the diet of the Saladoid culture from Puerto Rico. We also found evidence of the presence of bacteria known to inhabit the oral cavity of modern subjects in the Saladoid’s dental calculi, including Streptococcus sp., Porphyromonas sp., and Gemellaceae sp., supporting previous findings that bacterial members of the human oral microbiome are ubiquitous across diverse geographical regions and time periods (Adler et al., 2013; Warinner et al., 2014; Clemente et al., 2015).

Even though the DNA extraction and sequencing methods differed, our data are consistent with another study characterizing the dental calculi microbiome of a pre-Columbian Caribbean culture inhabiting the Virgin Islands, where Actinobacteria comprised a large proportion of the bacterial taxa (Ziesemer et al., 2015). This same study showed that tropical climates may not have the same degree of ancient bacterial DNA preservation as colder climates. Given that these samples have been in contact with soil for extended periods of time, it is expected to find 16S rRNA gene sequences matching soil microbiomes, even after external decontamination of the dental calculi; yet, one advantage of the bioinformatic tools developed for 16S rRNA gene analyses is that reads corresponding to soil contaminants can be filtered. Bioinformatic tools available to discriminate potential sources of contamination, such as SourceTracker, may not conclusively indicate that results in the present study were influenced by soil contamination. While bioinformatic analyses do not truly support that the presence of Actinobacteria is due to contamination with the post-mortem environment, it is feasible to hypothesize that the OTUs present in the dental calculi samples reflect, to some extent, dietary habits of the Saladoid culture. It is hypothesized that the Saladoid diet was mainly composed of root crops and fruits, whose residues are known to get entrapped in dental calculi (Mickleburgh & Pagán-Jiménez, 2012). It is also possible that intake of root crops and fruits contaminated with soil may have had some impact in the bacterial taxa being detected in the dental calculi samples. Differences in lifestyles and dietary habits may explain some of the lack of similarity between the Saladoid and ancient European oral microbiomes characterized previously, even when both of these ancient cultures lacked oral hygienic conditions. In addition, some members of the Actinobacteria, including Actinomyces have been associated with dental caries in modern individuals. Detection of Actinomyces DNA in the Saladoid is intriguing as it supports observations that pre-Columbian cultures from the Caribbean also suffered from oral diseases (Chalmers et al., 2015; Ziesemer et al., 2015). Our data may also be consistent with a previous study suggesting that dental caries in ancient human populations may have resulted from the intake of fermentable carbohydrates (Adler et al., 2013), as there is archaeological evidence suggesting that the Saladoid ingested fermentable carbohydrates (Widstrom et al., 1987; Humphrey et al., 2014) that are part of the root crops.

Results may have also been influenced by the amplified 16S rRNA gene variable region. Recent evidence suggests that amplification of different 16S rRNA gene variable regions may also have profound effects on ancient microbiome studies due to PCR amplification biases (Ziesemer et al., 2015). This study also suggested that amplification of the V4 region may be impractical for ancient DNA studies; however, <300 bp is still within what would be considered the norm in the field of ancient DNA (Drancourt et al., 1998; Von Wurmb-Schwark et al., 2004; Roberts & Ingham, 2008; Pääbo, 2012). In addition, the V4 region has been better characterized and has shown to have a better phylogenetic resolution (Yang, Wang & Qian, 2016) than other 16S rRNA gene variable regions.

Due to the small number of recovered sequences after filtering soil from the archeological site of Sorcé and blank control OTUs, special consideration should be taken when interpreting results from the prediction of functional categories in the dental calculi samples. In addition, while PICRUSt was utilized to predict functional profiles based on 16S rRNA gene data, which has shown to work more efficiently with stool microbiomes, results may still add valuable information regarding the potential effect of culture, lifestyle, dietary habits, oral hygiene and disease states to the ancestral state of human oral microbiomes. Ancient and modern oral microbiomes showed shared functional profiles, suggesting that certain predicted functional pathways may be ubiquitous to the human oral microbiome. While taphonomic conditions may have influenced the predicted functional profiles of the dental calculi included in the present study, it is also feasible to hypothesize that some of the predicted functional profiles were altered due to differences in culture, lifestyle, dietary habits, oral hygiene and disease states, which are known to affect the community structure and function of modern human oral microbiomes (Zarco, Vess & Ginsburg, 2012).

Conclusions

Skepticism is met when studying the oral microbiomes of ancient Caribbean cultures as there is always the risk that soil bacteria may have penetrated the dentine; thus, the reliability of the microbiome structure and function of dental calculi excavated from tropical regions may be challenged. Tools such as Bayesian SourceTracker analyses may be used to authenticate ancient microbiomes by providing insights into the contribution of possible contaminating sources, including soil from the archaeological site, as well as reagents. Our data may provide insights of the oral microbiome of the Saladoid culture from Sorcé, Puerto Rico, although a deeper sequencing depth is needed. Comparative studies would ideally include plaque and saliva samples from modern indigenous individuals, but access to these type of samples is challenging. Given that most human microbiomes studies have been performed with subjects from the US limits our ability to interpret some of the results. Bacteria present in the oral cavities of modern subjects were also present in the dental calculi microbiome of the Saladoid culture. Advantages and disadvantages exist when using 16S rRNA gene high-throughput sequencing to characterize ancient human oral microbiomes.

Supplemental Information

Figure S1 SourceTracker Filtered Unfiltered

SourceTracker analyses of the dental calculi samples including D21 (A), E19 (B), E26 (C), F20 (D), G18 (E), G21 (F), G22 (G), I19 (H), I23A (I), I23B (J), I24A (K), I24B (L), F24 (M), H6 (N) and M8 (O). Dental calculi microbiomes were compared to stool, coprolite, saliva, soil from the archaeological site of Sorcé, subgingival plaque, and supragingival plaque microbiomes. Dental calculi sequences not matching any of the microbiomes included were classified as unknown. Figure shows results prior and after filtering soil and blank control OTUs from the dental calculi.

Click here for additional data file.

Figure S2 Alphararefaction curves Filtered Unfiltered

Alphararefaction curves of chao1 (A), and observed OTUs values (B) after filtering soil and blank control OTUs from the dental calculi, and chao1 (C), and observed OTUs (D) values prior filtering soil and blank control OTUs. Samples included dental calculi from loose teeth samples (yellow), dental calculi from teeth attached to bone (red), supragingival plaque (pink), subgingival plaque (green), saliva (orange), coprolites (light blue), stool (blue), soil from the archaeological site of Sorcé (brown), and blank control (purple).

Click here for additional data file.

Figure S3 Alpha Diversity Unfiltered

Bar plots of alpha diversity indices. Bar plots representing the chao 1 (A) and observed OTUs (B) indices for the bacterial taxonomy based on 16S rRNA gene of the dental calculi, modern supragingival and subgingival plaque, saliva, coprolites, stool and soil from the archaeological site of Sorcé microbiomes. Alpha diversity indices were computed from the average of ten iterations using the collate_α.py workflow. Soil and blank control OTUs were not filtered from the dental calculi prior analyses.

Click here for additional data file.

Figure S4 PCoA Unfiltered

Principal Coordinates Analysis (PCoA) 2D plots of ancient and modern oral and gut microbiomes, as well as that of soil from the archaeological site of Sorcé. Dental calculi (yellow), dental calculi of teeth attached to bones that enabled the identification of gender or age (Dental calculi (Bone)) (red), coprolites (light blue), stool (dark blue), soil from the archaeological site of Sorcé (brown), supragingival (pink) and subgingival plaque (green), and saliva (orange). Soil and blank control OTUs were not filtered from the dental calculi prior analyses.

Click here for additional data file.

Figure S5 Taxa Plots Unfiltered

Barplots representing the bacterial taxonomy based on 16S rRNA gene. Data are shown at the phylum level for dental calculi, supragingival plaque, subgingival plaque, saliva, coprolites, stool, soil from the archeological site of Sorcé, and a blank control. Soil and blank control OTUs were not filtered from the dental calculi prior analyses.

Click here for additional data file.

Figure S6 PICRUSt Unfiltered

Heatmap of the relative abundances of the predicted functional categories (level 2) of microbiomes of dental calculi, coprolites, supragingival and subgingival plaque, saliva, stool and soil from the archeological site of Sorcé. Functional categories were predicted using PICRUSt. Soil and blank control OTUs were not filtered from the dental calculi prior analyses.

Click here for additional data file.

Figure S7 LefSe Unfiltered

Linear discriminatory analyses. Effect size (LEfSe) plots of predicted functional categories (level 2). Functional categories of dental calculi (ancient oral microbiomes, green), supragingival plaque, subgingival plaque and saliva (modern oral microbiomes, purple), coprolites (ancient gut microbiomes, red), stool (modern gut microbiomes, blue) and soil from the archeological site of Sorcé (tropical soil, light blue) were predicted using PICRUSt. Soil and blank control OTUs were not filtered from the dental calculi prior analyses.

Click here for additional data file.

Data S1 Data Set S1

OTU Table Blank Control

Click here for additional data file.

Data S2 Data Set S2

OTU Table Soil

Click here for additional data file.

Data S3 Data Set S3

Taxa found in blank control

Click here for additional data file.

Data S4 Data Set S4

Dental Calculi Genus Level Filtered

Click here for additional data file.

Data S5 Data Set S5

Dental Calculi Genus Level

Click here for additional data file.

Data S6 Data Set S6

Group Significance Dental Calculi Filtered

Click here for additional data file.

Data S7 Data Set S7

Group Significance Dental Calculi Unfiltered

Click here for additional data file.

Data S8 Data Set S8

Core OTUs 25% Dental Calculi

Click here for additional data file.

Data S9 Data Set S9

Core OTUs 25% Modern Oral Microbiome

Click here for additional data file.

Table S1 All Samples

Click here for additional data file.

Table S2 Chao1 Alpha Compare Filtered

Click here for additional data file.

Table S3 Observed OTUs Alpha Compare Filtered

Click here for additional data file.

Table S4 Chao1 Alpha Compare Unfiltered

Click here for additional data file.

Table S5 Observed OTUs Alpha Compare Unfiltered

Click here for additional data file.

We would like to acknowledge Mark Brolaski for providing reagents for the DNA extraction, and Christina Warinner for providing valuable insights in microbiome methods to study ancient dental calculi.

Additional Information and Declarations

Competing Interests

Author Contributions

Field Study Permissions

Data Availability

The authors declare there are no competing interests.

Tasha M. Santiago-Rodriguez conceived and designed the experiments, performed the experiments, analyzed the data, wrote the paper, prepared figures and/or tables, reviewed drafts of the paper.

Yvonne Narganes-Storde contributed reagents/materials/analysis tools, reviewed drafts of the paper.

Luis Chanlatte-Baik contributed reagents/materials/analysis tools.

Gary A. Toranzos wrote the paper, reviewed drafts of the paper.

Raul J. Cano conceived and designed the experiments, facility.

The following information was supplied relating to field study approvals (i.e., approving body and any reference numbers):

Permission to collect and process the samples was given by the Center for Archaeological Investigations at the University of Puerto Rico.

The following information was supplied regarding data availability:

NCBI BioProject PRJNA315748.

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
