# Peer review of "Insights of the dental calculi microbiome of pre-Columbian inhabitants from Puerto Rico"

_PeerJ, doi:10.7717/peerj.3277_

## Round 0.1 · original submission · Major Revisions

One major criticism raised by both reviewers which I fully agree with is the extremely low numbers of remaining sequences after rarefaction (only 245). Maybe the authors can reanalyze some of the samples (both experimentally and/or bioinformatically), if there are preamplification samples left. Another major valid point is a potential contamination by soil microbes. The authors should evaluate these points very critically, redo experiments, if possible, and adjust their conclusions. It is, by the data that they present, absolutely not excluded that all diversity that they see comes from a contamination. Of course the authors are expected to provide point-by-point responses to the reviewers' comments and address all the points adequately in a revised manuscript. The authors should critically ask themselves whether they are convinced of their present dataset. Before its publication in any revised form can be considered, major changes both in the experimental and bioinfomatical approach and data interpretations and discussion will be required, because of these very serious criticisms. It is not meant as a general criticism of the methodology, and it remains well understood and noted with due respect to the authors that providing meaningful data from ancient DNAs is a very serious challenge, even with the improvements in methodology that we have today.

Reviewer 1 ·

Basic reporting

The overall language is clear and professional. However, there are several instances where proper background and citations need to be included.

For example:
Line 63: The Caporaso et al. 2010 paper was the one that enabled high throughput 16S based microbiome characterization and is a more appropriate reference.

Line 73: The authors mention "Lifestyles, dietary habits and oral hygiene are also known to influence the oral microbiome", but no citations are included. In fact unlike the gut microbiome where there has been extensive work looking at bacterial communities in association with hygiene, diet and lifestyle, such studies are lacking for the oral microbiome.

Line 79: The Obregon-Tito et al. article was exclusively focused on the gut microbiome and not the oral microbiome.

Experimental design

The authors base their results on 16S V4 region characterization. Such amplicon based strategies have been shown to severely skew microbial community profiles (Ziesemer et al. 2015) as they are often targeting fragments well beyond the average fragment lengths of DNA preserved in ancient dental calculus.

The use of a commercial facility for building the 16S libraries is problematic. Ideally, the library build should also be performed in a dedicated ancient DNA facility.

The use of a commercial kit for DNA extraction is also problematic. There are specific protocols that have been developed for ancient DNA, aimed at recovering the extremely short fragments associated with these sources (median lengths often <100bp).

The authors mention two PCR reactions were used to enable amplification. Ideally, qPCR should be used to get relative 16S copy numbers to ensure that the extracts have endogenous bacterial DNA. Further, the authors do not mention how many cycles were used for the second PCR amplification. The authors also do not mention any use of decontaminated reagents (dNTPs, Primers, Polymerase) etc. which is the standard in ancient DNA work. These reagents are often the primary source of background DNA. Specifically, Polymerases often carry trace levels of bacterial DNA as they are often produced in bacterial expression systems.

Validity of the findings

The authors mention that the Saladoid calculus microbiome profile matches those reported from the Virgin islands by Ziesemer et al. However, the Ziesemer et al article specifically mentions that these samples had the poorest preservation with extensive soil contamination.

The authors have a final rarefaction depth of 245 sequences. They mention removing OTUs shared between blanks and soil controls. They should publish a table with the total number of sequences per sample (dental calculus, soil, and controls), and the number of sequences left after removal of those shared with soil and controls. In fact, microbial community comparison (beta diversity, sourcetracker etc.) should be performed using all the reads. By removing sequences shared with soil and blanks prior to these analyses, the authors are automatically reducing the ability to find any potential contamination.

The relative proportion of reads removed to reads retained will also provide a good estimate of contamination burden, and is essential before evaluating the authors claims.

Additionally, analyses should also be performed at the Genus level. Soil has extremely high bacterial diversity at the OTU level. However, at the Genus level, these will be collapsed, and allow for better estimation of soil contamination.

Reviewer 2 ·

Basic reporting

I believe the topic of ancient human microbiomes is very interesting as these micro-archaeological studies may contribute to our understanding of the current human microbiome.

Overall I believe the manuscript is a bit confuse in that its main focus (dental calculus) is somehow shifted as it compares also saliva from US HMP samples, stool samples from amazonians and coprolites.

Furthermore, it lacks a robust data analyses and a should review some of the references and statements.

Experimental design

The study design seems not to be a result of careful thought but rather a choice of samples by chance as they became available. It makes the interpretation of the resulting small dataset very confusing, as authors decided to compare with several different types of samples.

I suggest authors should focus only on oral microbiome and dental calculi and provide a possible change in community structure that could eventually be related to lifestyle or even diet. Although these assumptions are all very risky to take as only a couple of hundreds of sequences are analyzed by rarefaction.

Validity of the findings

Data analyses is not robust, lacking information on phred scores, sequence size, amount of chloroplast chimera removal and singletons. Rarefaction was extremely low (possibly to avoid loosing samples, even though sequence numbers were higher). Rarefaction was n=245 seqs, from a dataset of thousands of sequences according to table 1. Data analyses must be carefully evaluated.

This has great impacts into the discussion assumptions that the structure and function of the calculli microbiota may reflect a given horticulturalist lifestyle or dietary choices. This is a great stretch and renders the manuscript in the current form rather weak.

Additional comments

1-lines 71-76 - does not make to talk about dysbiosis when this MS is about an ancient culture where no “normal” microbiome is known.

2-lines 95-97- It is feasible to hypothesize that the recovered dental calculi may resemble taxonomic and functional profiles of modern dental plaque, and also possibly reflect pre-Columbian lifestyles. This does not make sense has the 3D beta diversity plots show that there are great differences in dental calculi and modern oral microbiota. I suggest a core microbiota to be assessed from these two sample types.

3-line 100 due to knowledge in seafaring.(needs reference)

4-line 152-Please describe in detail methods to which you determined age-at-death and sex estimates.

5-line 153 - specific regulations for biospecimens? Please specify

6-line 165 - deal scalar (please add company, state)

7-line 190 - references to 16S V4 primers have a mistaken reference! those are Caporaso JG, Lauber CL, Walters WA, Berg-Lyons D, Huntley J, Fierer N, Owens SM, Betley J, Fraser L, Bauer M, Gormley N, Gilbert JA, Smith G, Knight R. 2012. Ultra-high-throughput microbial community analysis on the Illumina HiSeq and MiSeq platforms. ISME J. Please add this one instead.

8-lines 211-216 - Methodology for 16S rDNA data analyses lacks information on chimeras, chloroplast and singleton removal, Phred score and sequence size.

9-Table 2 - first column says "OUT" instead of OTU

10-Rarefaction level is very low n=245 seqs, from a dataset of thousands of sequences according to table 1. Table needs to have the number of OTUs.

11-Beta diversity plots have undergone quite a modification, I suggest a weighted 2D plot as the variance is not well explained through any of the axis.

12-PICRUSt inferences on such a low amount of data are quite overestimated and Picrust results on Kegg profiles do not seem to change from stool, saliva or supra gingival teeth. I believe these assumptions are too exxagerated as they are based on a very slim dataset.

---

## Round 0.2 · accepted · Accept

Dear authors, dear Tasha,

We are pleased to accept your manuscript for publication. The paper has been extremely thoroughly revised according to reviewers' useful suggestions. The authors have probably noted a recent Nature paper about ancient dental calculus microbiome (Weyrich LS et al., Nature March 2017). It should be encouraged if the authors would like to cite and reference this article as information in proof stage - no inhibition if you think one or two sentences comparing your data to those published in the other article would be useful and will enhance the visibility of your article.

Congratulations and best regards,
Christine Josenhans
* * *
Academic Editor PeerJ